# Detection of the Complete ECG Waveform with Woven Textile Electrodes

**DOI:** 10.3390/bios11090331

**Published:** 2021-09-13

**Authors:** Katya Arquilla, Laura Devendorf, Andrea K. Webb, Allison P. Anderson

**Affiliations:** 1Smead Aerospace Engineering Sciences, University of Colorado Boulder, Bouder, CO 80303, USA; apanders@colorado.edu; 2Department of Information Science, University of Colorado Boulder, Bouder, CO 80303, USA; laura.devendorf@colorado.edu; 3The Charles Stark Draper Laboratory, Inc., Cambridge, MA 02139, USA; awebb@draper.com

**Keywords:** e-textiles, wearable sensor systems, smart garments, textile electrodes

## Abstract

Wearable physiological monitoring systems are becoming increasingly prevalent in the push toward autonomous health monitoring and offer new modalities for playful and purposeful interaction within human computer interaction (HCI). Sensing systems that can be integrated into garments and, therefore, daily activities offer promising pathways toward ubiquitous integration. The electrocardiogram (ECG) signal is commonly monitored in healthcare and is increasingly utilized as a method of determining emotional and psychological state; however, the complete ECG waveform with the P, Q, R, S, and T peaks is not commonly used, due to the challenges associated with collecting the full waveform with wearable systems. We present woven textile electrodes as an option for garment-integrated ECG monitoring systems that are capable of capturing the complete ECG waveform. In this work, we present the changes in the peak detection performance caused by different sizes, patterns, and thread types with data from 10 human participants. These testing results provide empirically-derived guidelines for future woven textile electrodes, present a path forward for assessing design decisions, and highlight the importance of testing novel wearable sensor systems with more than a single individual.

## 1. Introduction

Biosignal monitoring through wearable systems is essential for the growing area of remote healthcare and is emerging as a pillar of human-computer interaction (HCI), thereby, enhancing the fluency of communication between humans and the computational systems they interact with daily. The ubiquity of biosignal monitoring is driven by the advancement and availability of wearable sensor systems. Industry has developed sleek, user-friendly systems that are far more attractive than their laboratory-grade counterparts; however, they often do not provide raw data, only producing aggregate metrics like heart rate (HR) that miss more intricate variability.

In the electrocardiogram (ECG) signal, often only the most prominent R peak is detected to understand heart rate variability (HRV). The smaller peaks within the ECG waveform (the P, Q, S, and T peaks) are used in clinical practice to detect cardiac illness; however, the complete ECG waveform is rarely detectable with wearable systems. Complementary research shows that this more fine-grained and intricate variability data is necessary for purposes of measuring emotional or stress responses, which are key aspects of HCI [1,2]. Given this need, the focus of this work is the design, development, and performance evaluation of woven textile electrodes with the goal of reliably detecting the full ECG waveform.

## 2. Background: Textile-Integrated Electrodes

A high-quality ECG signal can only be achieved with electrodes that are correctly placed, maintain contact with the skin without slipping, and pick up the small electrical signals coming from the heart without great impedance. Unlike physicians, designers of novel wearable systems must balance the production of high-quality signals with comfort and acceptability.

Single-use adhesive electrodes are the standard for ECG monitoring in laboratories (shown in Figure 1). These electrodes typically consist of an adhesive pad where one side sticks to the skin and the other side provides an easy attachment point for an electrical connection. While adhesive electrodes provide high-quality data, they can cause skin irritation and are uncomfortable for long-term (longer than an hour) use. Furthermore, the conductive gel used within adhesive electrodes dries out over time, causing temporal changes in signal quality, which renders them unsuitable for pervasive or long-term monitoring [3].

These findings have led researchers to create “dry” electrodes—electrodes without any conductive gel—as a potential improvement to the current standard. The comfort advantages offered by dry electrodes have led them to be used by many commercially-available systems such as the AppleWatch (Apple, Inc., Cupertino, CA, USA) and Fitbit (Fitbit, Inc., San Francisco, CA, USA). These products are commonly used among the general public and provide HR information in a user-friendly way by sampling the ECG signal at one point on the body (e.g., the wrist).

Having more connection points, or multiple-lead monitoring, is important for capturing the intricacies of the ECG waveform that allow the measure of heart rate variability (HRV)—a measure of variability between each heart beat instead of an aggregate average. The Empatica watch (Empatica, Inc., Milan, IT) and Hexoskin (Carre Technologies, Inc., Montreal, QC, CA) are two of the more capable commercially-available systems used often for research; however, the data quality of the Empatica is variable and significantly impacted by wearer movement, and the Hexoskin is custom-fit for each individual, making it cost-prohibitive.

To address the challenges around comfort, wearer movement, and connection points, many have turned to textile-based dry electrode designs and the development of dry textile electrodes that improve upon currently commercially available systems. Researchers have developed and incorporated these electrodes into garments through a multitude of different methods utilizing conductive threads [4,5,6], yarns [7,8,9], pre-fabricated textiles [10,11], pastes [12], and inks [13,14,15,16,17,18,19,20].

These studies employ a variety of different methods for integrating conductive elements. Researchers developed some electrodes by knitting conductive yarns together; however, the stretchy nature of knit fabrics is not ideal for electrodes that need to maintain constant resistance [7]. Others coated preexisting fabrics with conductive inks and pastes, requiring advanced manufacturing methods and/or suppliers that can be expensive in finance and time [12,13].

Those developing sewn electrodes showed that they are cost-effective but do not allow much design freedom beyond stitch pattern [4]. While many of these studies presented promising results, the electrodes were not often tested on more than a single subject, limiting the ability to determine whether these electrodes will be functional on diverse groups of individuals with different skin characteristics (e.g., dry or moist and hairy or smooth).

Woven electrodes were chosen for development because conductive elements can be integrated seamlessly (and are invisible to both the wearer and the observer) and exist in stable, inelastic woven textile structures that will not change in resistance. Weaving is a process by which sets of parallel yarns in two directions (warp and weft) are interlaced together into various patterns and mechanical structures. Our work is perhaps most similar to the work of Song et al., who, in 2010, demonstrated the design and efficacy of woven electrodes using similar traditional weaving methods [21].

The electrodes developed in that work were capable of producing recognizable ECG signals using the same data acquisition system and interface used in this work, but there are subtle differences in design that make their approach complementary to our own. The electrodes developed by Song et al. were of one size, and the two different types were “flat” and “convex”—the threads within the convex electrodes were left relatively loose and filled with a conductive paste.

The main point of comparison for electrode types was the signal-to-noise ratio (SNR), and thus this analysis adds to the tools available for comparing these types of woven electrodes with the current work by adding information about electrode size, patterning, and design constraints. Furthermore, these designs were tested without the use of conductive paste, as this is seen to be a hindrance to feasibility and wearability.

## 3. Materials and Methods

### 3.1. Equipment

While knits are composed of intersecting loops, woven fabrics are created by interlacing yarns in perpendicular directions with the support of a loom. The loom used for the development of these electrodes was a TC2 digital jacquard loom (Tronrud Engineering, Norway). The loom holds one set of threads taut in the “warp” direction, and the “weft” thread is mounted on a shuttle and thrown through the gap in the warp threads (called the “shed”) as one continuous thread. Patterns are created by lifting different combinations of warp threads as the shuttle with the weft thread is thrown through, and each thread is lifted by a carrier called a “heddle”.

Non-jacquard style looms have a number of different pattern options that must be configured before weaving begins, but the TC2 loom has a single individually controllable heddle for each thread. This means that each of its 1320 warp threads can be lifted independently of one another, opening up design opportunities beyond repeating patterns. The lifting of each heddle is controlled by a computer program that reads bitmap images, and the weaver’s part in the weaving process consists of pressing the pedal to advance to the next row of the bitmap image and throwing the shuttle.

Conductive elements can be added into the textile by weaving with multiple shuttles. Specifically, while the base yarn that holds the supporting structure may travel the entire width of the fabric, individual and continuous strands of conductive yarns can be integrated into specific areas using an “inlay” technique (described in more detail in [22]). This is the process that was followed to develop the woven electrodes described herein. All samples were woven at 30 ends-per-inch (meaning there are 30 equally spaced yarns within every inch of the yarns’ width).

### 3.2. Materials

Two different conductive threads were chosen for comparison: steel (Sparkfun Electronics, Boulder, CO, USA) and silver-coated nylon (Statex, Bremen, DE) (shown in Figure 2). These were chosen because the are both accessible forms of conductive yarns and differ in their structure and price point. The steel thread is stranded steel spun into a thread—there is no traditional nylon or cotton thread at the core. This resembles a structure of a spun staple fiber yarn wherein individual short lengths of steel fiber are given structure through the application of spin or twist. Vogl et al. noted in their study of stretch sensors that this structure compresses fibers upon press or stretch, thus, changing the values of resistance [23].

Namely, when a spun staple fiber yarn is compressed, there are more connection points between the individual staple fibers and, thus, a shorter path for current to travel from power to ground, and therefore a lower resistance value than when the yarns are at rest. The nature of the spun fibers, and stainless steel fibers in particular, leads to this thread feeling scratchier than other conductive threads. It can also be said to be “grabbier” when woven, meaning that its loose ends have the capacity to grab onto neighboring threads, adding friction.

The silver-coated thread consists of several filaments of nylon, which are coated in silver nanoparticles and spun together. This is a multi-filament type yarn, whereby each individual filament stretches, reducing the cross-sectional area and, thereby, offering a thinner channel for current travel, causing more resistance. This thread is more conductive than the steel thread but is more expensive and delicate.

The conductive yarns are held in their structures by supporting non-conductive yarns in both the warp and the weft directions of the fabric. All supporting base yarns are 20/2 mercerized cotton thread, in black (warp) and white (weft). Cotton was chosen to prevent the woven structure from being severely impacted by a more textured thread or a material with strong variability based on moisture absorption, such as wool.

### 3.3. Electrode Design

Four different weave structures were used for the design of these electrodes: 1/15 sateen, 1/15 twill, 1/15 broken twill, and 1/15 birdseye structure. These structures were chosen in consultation with Sandra Wirtanen, a professional weaver. Despite the endless patterning possibilities with the TC2 loom, these four patterns were tested as a first step toward understanding the functionality of woven electrodes to provide a solid foundation from which deviations and explorations can continue in the future. These four patterns were selected for electrode development based on the desire to maximize the contact area between the conductive thread and the skin and to minimize movement within the electrode that can create noise.

In any woven structure, a weft thread must be periodically anchored by a warp thread, and changing how often that anchoring occurs changes the weave structure. The number of anchors (or what are also called interlacements) across the width of the weft determines how much the weft yarn is able to “float”, or curve between anchor/interlacement points. The direction of the curve (e.g., along the surface vs. protruding out from the surface), is determined by the stiffness of the materials composing the design, as well as the structure overall.

This ability for yarns to float in a structure is one way to make textiles that are not simply flat 2D surfaces, but add textures and “puffiness” to the surface. This puffiness was varied as a technique to increase/decrease skin contact. If the weft thread is anchored too infrequently, there will be long “floats” (loose threads) that can move excessively within the electrode and cause noise in the electrode. If the thread is anchored at every other warp, as it is within the plain weave structure, it will not float at all, and will instead stay rigidly in place within its row. Figure 3 shows the difference between the most anchored and least anchored patterns.

The four patterns tested are of the 1/15 variety, meaning they anchor the yarn once every 15 yarns (or once every 1/2 inch, because the loom is warped at 30 ends per inch). Yet, the four patterns varied in how they anchor the yarns in subsequent rows. For every electrode, the conductive yarn was integrated as a supplemental weft, in addition to the cotton weft that spans the width of the fabric. This added additional volume and helped us maintain a consistent structure across the width of the electrode.

The 1/15 sateen (Figure 3a) maximizes skin contact with the conductive threads with the lowest anchoring points. Yet, the length of the floats are prone to snagging and tend to hang down the surface of the fabric. The 1/15 broken twill (Figure 3b) has moderate puffiness and space between anchors. This created a more recognizable zig-zag pattern in the resulting electrdoe by aligning the floats in shifting diagonal sections. This structure is less prone to snagging and is more rigid than the sateen.

The 1/15 twill (Figure 3c) has moderate puffiness and space between anchors, similar to the broken twill. This created more recognizable and compact diagonal ribs by aligning the floats in shifting diagonal sections. This structure is less prone to snagging and is more rigid than the sateen. The 1/15 birdseye (Figure 3d) has the least space between anchors and, thus, the least skin contact. Yet, its pattern is the most crisp and recognizable. This structure is less prone to snagging and is more rigid than the sateen.

The weave structure, as well as the yarn type, impacts the profile of the electrode and how it feels against the skin. Designers will need to consider which structures maximize comfort to encourage adoption of the system. Perception of the electrodes within garments is also impacted by the woven structure. Users may find the cleaner-looking structures, such as the birdseye pattern more appealing than the sateen.

With the goal of investigating the impact of electrode size on signal quality, a set of eight electrode types with sizes ranging from 2.54 × 2.54 cm (1 × 1 inch) to 8.46 × 8.46 cm (3.33 × 3.33 inch) was designed. Smaller electrodes have fewer rows of thread within them, and thus they have the potential to produce less noise due to movement; however, they have less contact area on the skin, which can impact the signal strength. Increasing the electrode size allows a greater contact area but introduces the issue of conforming to body curvature.

Gaps between the electrode and the body can occur, impacting the consistency of skin contact and ultimately creating noise within the data. The goal in exploring this range of sizes was to see trends in these effects and identify an optimal size for future designs. These eight electrodes used to compare size were woven with the 1/15 sateen pattern and steel thread only, to eliminate those variables from the size comparison. These results are detailed in Arquilla et al., 2020, and therefore they will not be repeated herein [24].

Devendorf et al. developed the AdaCAD software to improve the process of designing woven textiles with conductive elements, and they incorporated procedures for developing these types of woven electrodes [25]. This software will enable the production of more intricate and well-integrated electrode designs in the future. A tutorial and further resources can be found at http://adacad.unstable.design/electrodes/ (accessed on 11 September 2021).

### 3.4. Experimental Design

For data acquisition, electrodes were hard-wired to a BIOPAC MP160 data acquisition module (BIOPAC Systems, Inc., Goleta, CA, USA). The BIOPAC is an FDA-approved system for monitoring the ECG signal, and thus it is often used to validate the efficacy of new wearable technologies and peak detection algorithms [26]. Before testing, a power analysis was performed to determine the number of participants needed to test to see a 10% change in signal quality. This calculation produced eight participants, so that number was increased to ten to account for potential attrition.

The recruitment of users presented some challenges as each participant had to have relatively consistent weights and musculature and also needed to apply our electrodes to the bare chest for testing. Since the quality of garment fit depends largely on body fat and tissue at the point the electrode contacts the skin, including participants with breasts would have required the development of additional mounting garments.

Thus, 10 male-identified participants whose weights ranged between 130 and 170 lbs were recruited. All participants were between the ages of 21 and 30. While the bias of our participant recruitment and testing setup is acknowledged in our limitations section, major impacts on the results are not anticipated. This restricted selection did allow at least the validation of the baseline claim of whether the woven electrodes could provide comparable data to adhesive electrodes and the particular design guidelines that impact the quality.

These recruitment restrictions were also implemented to allow the design and use of a single adjustable garment to house the electrodes on all participants. Each participant was tested with one set of adhesive electrodes and the 16 sets of woven electrodes in a within-participants design to account for inter-individual variability. Participants were instructed to remain seated, relaxed, and as still as possible during data collection. Additional cloth straps tied on the outside of the integrated garment to exert pressure on the electrodes were also used in an effort to maintain consistent skin contact between participants.

While we did attempt to maintain the same pressure for each participant, this process was conducted without a pressure sensor; therefore, we cannot, with certainty, claim that all electrodes were tested at the same pressure for each participant. We acknowledge this potential limitation of the study. While the goal of these electrodes is to eventually integrate them into garments worn during a variety of activities, this first assessment was focused on understanding the properties of different design characteristics, and, for that, it is important to remove other sources of variability (such as movement) to reveal the subtle differences due to design.

## 4. Evaluation Methods

### 4.1. Signal Processing

The R peak is the most distinct and essential feature of the ECG signal; it is the most pronounced feature and is used to calculate R-R interval (time between beats) and heart rate (HR; usually measured in beats per minute). The time intervals between the R peaks are used to assess heart rate variability (HRV), which captures the beat-to-beat variability that cannot be captured with a simple HR measurement.

Researchers can measure acute changes using this beat-to-beat variability, while the HR produces information about physical and psychological health in broader strokes that may be less rich for the time scales required to support real-time monitoring. Wearable ECG monitoring systems do not often produce data of high enough quality to detect the small peaks within an ECG waveform, and thus the analysis of the detectability of these small peaks is an important and novel investigation for these woven electrodes.

### 4.2. Data Analysis and Comparison Metrics

The detection of the R peak and the smaller peaks within the waveform (P, Q, S, and T) through an automated algorithm were compared with peak detection using visual inspection. Data from each test were exported and processed in Python using the neurokit2 toolbox. The data were all filtered using a Butterworth filter. After filtering, the best 10 s of each sample were chosen for this analysis. The commonly used precision, recall, and F1 metrics were used to conduct comparisons between the weave structures, electrode sizes, and thread types.

## 5. Results

### 5.1. Weave Structures

The twill and birdseye patterns showed the highest median performance metric values with the tightest distributions (shown in Figure 4). These two patterns have higher anchoring and lower skin contact than the other two, so this difference in scores indicates that anchoring the threads to avoid movement within the electrode may be more important than maximizing skin contact when detecting all peaks within the waveform.

### 5.2. Thread Type

Differences in the quality of measurement in all structures, across both materials, were observed to be strongly correlated with the subject. Put another way, some subjects’ data were noisier or cleaner overall. This may have to do with how sweaty/dry the participant’s skin was, as more moisture on the skin would increase conductivity and correlate with better detection rates. We did not measure sweat accumulation during testing; therefore, we do not know what impact sweat has on the signal quality. Both sets of electrodes performed similarly for the R and S peaks; however, the silver electrodes performed better (with higher median values and tighter distributions) in detection of the P, Q, and T peaks (shown in Figure 5).

At the small electrode sizes tested, the staple fiber structure of the steel may have been more sensitive to noise caused by small movements. Furthermore, it is also possible that the signal was also picking up EMG signals while collecting the ECG. This could indicate that the silver electrodes were less prone to picking up noise that can cause false positives in R peak detection. The non-conductive nylon core of the silver thread could act to stabilize the electrode against picking up noise and ultimately reducing the false positive rate. As the entire steel thread consisted of conductive fibers, it is easier for movement within the electrode to cause small shifts in resistance that can produce noise.

### 5.3. Comfort

A comfort survey was used to assess the differences between the woven electrodes as a group and the adhesive electrodes. Differences in comfort between each pattern and size of woven electrode were not tested to avoid survey fatigue (this would have required 17 separate survey administrations). Instead, participants were asked to complete the comfort survey once after measurement with the adhesive electrodes was complete and once after all measurements with the woven electrodes were complete.

The survey consisted of 14 words describing potential sensations caused by the electrodes (some positive and some negative) that each subject rated in severity on a Likert scale from 0 to 4, with 4 being the most severe. The words that had the most changed median responses between the adhesive and woven electrodes were “loose”, “lightweight”, “stiff”, “sticky”, “non-absorbent”, “cold”, “clingy”, “rough”, and “scratchy”. The responses for the scratchiness of the woven electrodes were higher in median and in range compared with the adhesive electrodes. The woven electrodes were rated as less clingy, cold, and sticky compared with the adhesive electrodes, showing the difference made by using electrodes mimicking clothing. Median values for each sensation are shown in Table 1.

### 5.4. Inter-Individual Variability

The data from the 10 different participants showed a high degree of inter-individual variability. In the performance metrics for each participant, there was a clear difference in the performance metrics based on the individual being tested. For example, participant 9 had relatively high performance metrics for all peaks within the waveform with all of the textile electrodes. Figure 6 shows the ECG signals from participants 9 and 10 with the adhesive electrodes and one set of woven electrodes. There is a clear difference in the signal quality—as seen through a visual inspection of these signals—between the participants when tested with the woven electrodes; this difference is also reflected in the calculated performance metrics. These differences could be due to inherent skin moisture or perhaps the fit of the test garment on each participant.

### 5.5. Washability

For all functional garments, it is important to assess whether washing them impacts their performance. While extensive wash testing was not conducted with these woven electrodes, a single wash cycle did not produce any degradation. Other groups have investigated the impact of washing on conductive threads [27], and one group conducted wash testing with the same silvercoated thread used here and found minimal changes in resistance after eight wash cycles, indicating that signal quality would be unchanged [4].

## 6. Discussion

These experimental results demonstrate that woven electrodes are a viable option for collecting ECG signals. They also show that design decisions can have measurable impacts on signal quality, and thus it is important to quantify the signal quality when designing garment-integrated electrodes for physiological monitoring. Our findings show that silver thread may be less prone than steel thread to detecting false positives, which is a benefit for using these electrodes in conjunction with an automated detection system. Comments from participants and other members of the lab also indicated that the silver thread was less scratchy than the steel thread, which is important for maintaining comfort.

Comparisons of the detection rate and performance metric also revealed that the broken twill patterns may not be as successful as the other patterns using both thread types. Lastly, sizes 6 and 7 appeared to be the best for ECG signal detection, and size 1 was clearly too small to perform well. For future designs, working with electrodes close to sizes 6 and 7 is recommended.

Reflecting on the data overall, the silver thread integrated into the 1/15 sateen structure provided data that were only slightly noisier than the data collected with adhesive electrodes on most subjects and were able to produce the smaller parts of the ECG waveform that allow the measurement of HRV. More differences between the patterns were anticipated, and one possibility for the small differences has to do with the size at which the patterns were tested. Testing larger sizes of each pattern may reveal more variability between the patterns.

The increase in variability for scratchiness suggests that some participants found the woven electrodes scratchy while others did not, but there was widespread agreement that the adhesive electrodes were not scratchy. The differences in the clingy, cold, and sticky descriptors are a clear benefit of the woven electrodes as compared to their traditional counterparts. However, the lack of adhesive requires a tighter garment to maintain skin contact; thus, it is no surprise that the woven electrodes scored higher on average in snugness when compared with the adhesive electrodes.

Inter-individual variability is an important factor to consider in the design of wearable sensor systems that would not have been revealed had these electrodes not been tested with a group of 10 participants. This multi-user testing, even with a small group, is not often done in the development of wearable sensor systems, but this work demonstrates that it should be by showing that the specific participant does impact the performance of the electrodes. Studies with multiple participants tested are sparse in the literature, and there is not a good library of information on what characteristics cause these differences. Systematic testing with multiple participants across research groups will augment the quality of our understanding regarding wearable electrode performance.

### Limitations

While the electrodes were tested on 10 different participants, no female-identified participants or those who did not have the specific body type qualities stated were tested. This study also lacked the capacity to test on a large number of participants (n > 30) for this first development effort. Increasing sample size and diversity in future efforts will further inform design decisions. Additionally, wiring and electronics have not yet been integrated into the woven system, and therefore a standalone system is not yet test-ready. This is currently an area of research in our group.

## 7. Conclusions

This paper presented the design, development, and testing of eight different sizes and four different patterns of woven textile ECG electrodes in two different thread types. Each set of electrodes was tested with 10 different participants, exploring the differences in performance due to electrode size and discovering differences due to inter-individual variability. The standardized process of data analysis used in this work will continue to be used to quantify the textile electrode performance and make these results transferable to the design process.

Given the importance of maintaining skin contact for reducing motion artifacts within the data, our current work is focused on integrating the woven electrodes into a garment designed specifically for the female form. Garment fit is essential for high-quality data collection and will continue to be a focus of our future work.

## Figures and Tables

**Figure 1 biosensors-11-00331-f001:**
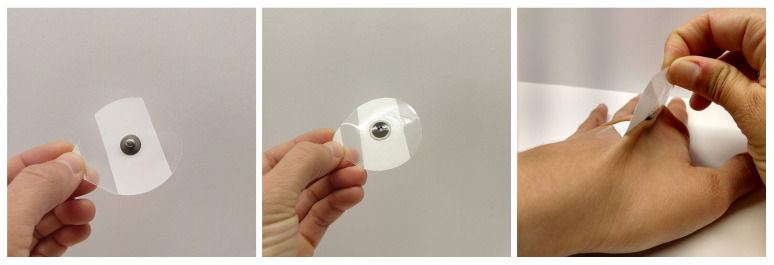
Front and back of a traditional electrode and the skin discomfort caused by the adhesive backing.

**Figure 2 biosensors-11-00331-f002:**
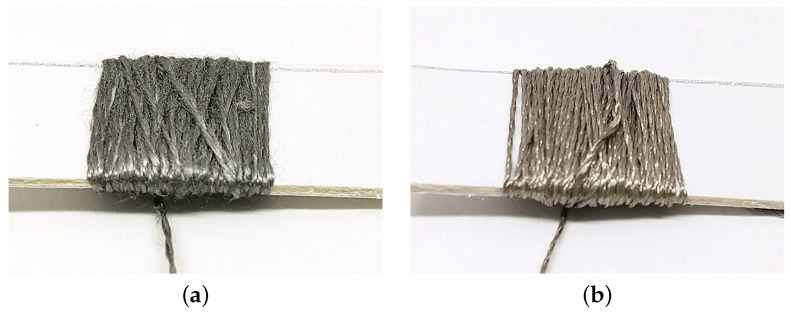
(**a**) Spun steel staple fiber thread. (**b**) Silver coated multi-filament thread.

**Figure 3 biosensors-11-00331-f003:**
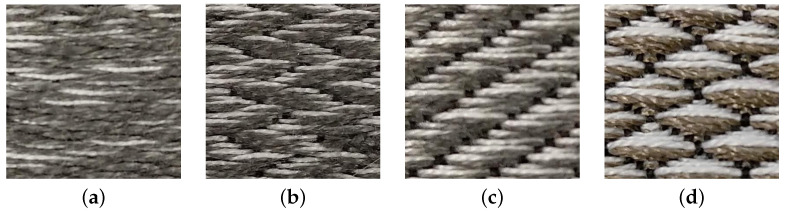
(**a**) Sateen: lowest anchoring, highest skin contact with the conductive portion of the electrode. (**b**) Broken twill: medium anchoring and skin contact. (**c**) Twill: medium anchoring and skin contact. (**d**) Birdseye: highest anchoring, lowest skin contact.

**Figure 4 biosensors-11-00331-f004:**
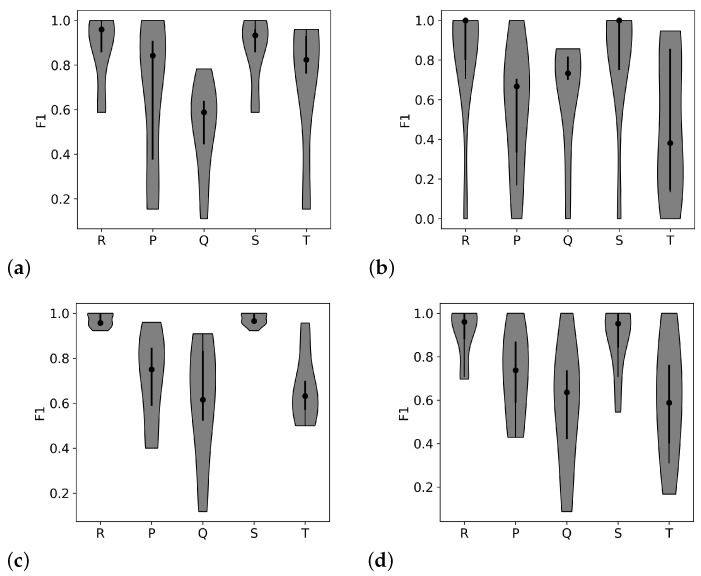
Each plot shows the F1 metric (the harmonic mean of precision and recall) for the four different weave structures with silver threads, tested with all participants. The x-axis labels the peak within the ECG waveform. The dot shows the median value, the black line shows the interquartile range, and the shaded area shows the shape of the distribution of scores. Higher median values and tighter distributions indicate better performance. The patterns shown are as follows, starting from the upper left corner and continuing clockwise: (**a**) sateen, (**b**) broken twill, (**c**) twill, and (**d**) birdseye. The twill pattern with medium anchoring showed the highest performance for R peak detection and the detection of the smaller peaks.

**Figure 5 biosensors-11-00331-f005:**
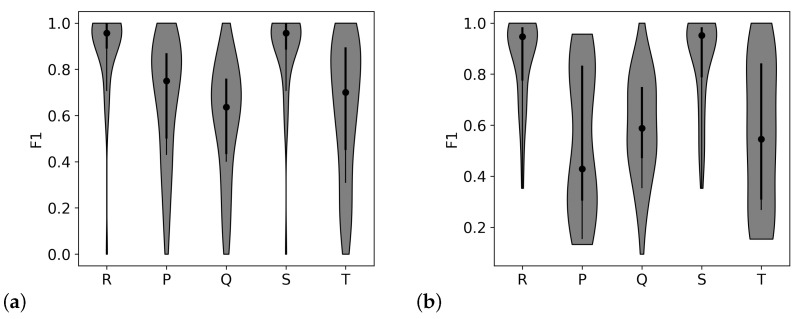
Each plot again shows the F1 metric for each thread subset of electrodes. The left plot (**a**) shows the data from all electrodes with silver thread across all participants, and the right plot (**b**) shows the data from all electrodes with steel thread across all participants. Both sets performed similarly for the R and S peaks; however, the silver electrodes performed better (with higher median values and tighter distributions) in detection of the P, Q, and T peaks.

**Figure 6 biosensors-11-00331-f006:**
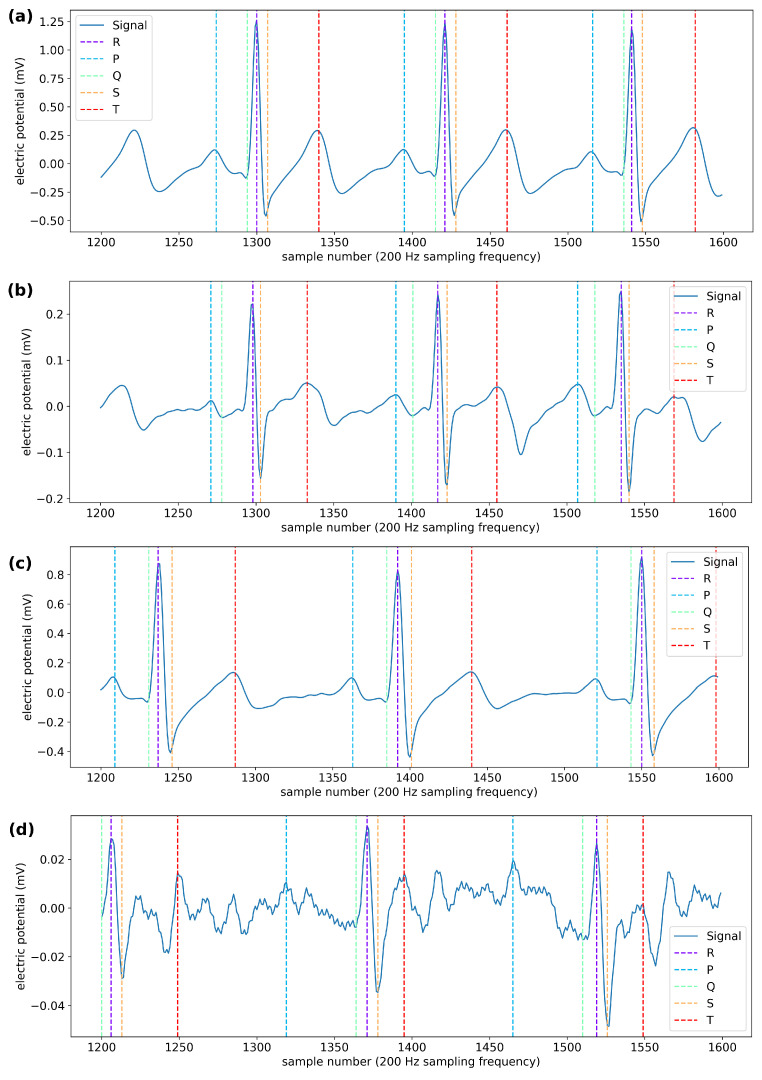
Each plot shows the ECG signal with each peak marked. (**a**) Signal from participant 9 using the adhesive electrodes. (**b**) Signal from participant 9 using the size 6 woven electrodes; the full ECG signal is not corrupted with noise, and most of the peaks are correctly detected. (**c**) Signal from participant 10 using the adhesive electrodes. (**d**) Signal from participant 10 using the same size 6 woven electrodes; the signal from participant 10 contains more high frequency noise and more large-scale noise features. Given the relatively clean adhesive signals for both participants, this suggests variability in the interaction between each individual and the woven electrode system.

**Table 1 biosensors-11-00331-t001:** Median comfort scores for each sensation with the adhesive and woven electrodes. Higher values indicate a low presence of the sensation, and lower values indicate a high presence of that sensation.

Sensation	Adhesive	Woven
snug	2	2
loose	4	3.5
heavy	4	4
lightweight	1	2
stiff	4	3
static-y	4	4
sticky	3.5	4
non-absorbent	3	2
cold	3	4
clammy	4	4
damp	4	4
clingy	3	4
rough	4	3
scratchy	4	3

## Data Availability

The experimental data set will be made available on Physionet.org.

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
