# Peer review of "Detection of the Complete ECG Waveform with Woven Textile Electrodes"

_biosensors, 2021, doi:10.3390/bios11090331_

Round 1

Reviewer 1 Report

  1. The aim of this article is to figure out the most suitable woven textile for the complete ECG morphology detection. I found out that it is well-organized paper.
  2. In the introduction, you mentioned that the absent of studies with diverse groups of individuals with different skin characteristics. So can you give us more about the subjects? What I found out are only the weights of them. I think it might be clue why there is inter-individual variability
  3. How did you control the pressure on the electrode? Are you sure that all measurements are in the same pressure?
  4. In the Figure 6, it would be better if each plot has a title as (a) ~ (d). The vertical lines that are named 0~4 do not clear what they mean. Did you mean P, Q, R, S, and T peak? If so, it would be better their names than just numbers for the readers.
  5. For autonomous health monitoring, a robust strategy to eliminate motion artifacts should be considered. I hope your further study is about on the woven textile electrodes for minimizing the motion artifact.
  6. Typos
  • At the top of the first page, “University of Colorado BOulder” is should be “University of Colorado Boulder”
  • In the line number 213, what is the meaning of “[88-kwon, 125-ragot].”
  • In the line number 242, there is an unnecessary vertical bar.

Author Response

We thank both reviewers for the time taken to provide helpful feedback on our manuscript.  The changes we have made in the manuscript are highlighted in red, and our response to each comment is provided in the italicized text after each numbered item.

Reviewer 1

  1. In the introduction, you mentioned that the absent of studies with diverse groups of individuals with different skin characteristics. So can you give us more about the subjects? What I found out are only the weights of them. I think it might be clue why there is inter-individual variability

Thank you for this comment.  We have added information on the age range of the participants, but we did not specifically measure skin characteristics, so we cannot provide that information for this experiment.  We agree that this type of measurement would be valuable in the future. 

  1. How did you control the pressure on the electrode? Are you sure that all measurements are in the same pressure?

Thank you for this comment.  We’ve added the following at line 235 to explain this further: “While we did attempt to maintain the same pressure for each participant, this process was conducted without a pressure sensor, so we cannot with certainty claim that all electrodes were tested at the same pressure for each participant.  We acknowledge this potential limitation of the study.”

  1. In the Figure 6, it would be better if each plot has a title as (a) ~ (d). The vertical lines that are named 0~4 do not clear what they mean. Did you mean P, Q, R, S, and T peak? If so, it would be better their names than just numbers for the readers.

Thank you for these suggestions.  We have made all of these changes to Figure 6.

  1. For autonomous health monitoring, a robust strategy to eliminate motion artifacts should be considered. I hope your further study is about on the woven textile electrodes for minimizing the motion artifact.

Thanks for this comment, we completely agree and have added a statement describing our current efforts in this area to the conclusion section.

  1. Typos
  • At the top of the first page, “University of Colorado BOulder” is should be “University of Colorado Boulder”

Thanks for catching this, the error has been fixed in the manuscript.

  • In the line number 213, what is the meaning of “[88-kwon, 125-ragot].”

Thank you for catching this too, the error has now been fixed.

  • In the line number 242, there is an unnecessary vertical bar.

Thanks for reading so closely, we’ve fixed this too.

Reviewer 2 Report

In this paper, the author presents a woven textile electrode as an option for an ECG monitoring system to capture the ECG waveform. After tested different patterns and threads, the author got some interesting results. Generally, this paper is well organized, but I have some questions here.

1) Some writing typo like page 3, line 123 "traditional."

2) Please change Figure 1 and using your device image to replace it.

3) How did the author obtain table 1? Is it just a  survey for the volunteer? If so, I do not think this table is appropriate since the sample size is too small.

4) Did the author consider the effect of sweat (will sweat affect the conductivity)? 

5) It will be more convincible if the author can provide a plot figure (similar to Figure 6) to evaluate the washability (before and after the wash). Also, a microscope image before and after washing is necessary.

Author Response

We thank both reviewers for the time taken to provide helpful feedback on our manuscript.  The changes we have made in the manuscript are highlighted in red, and our response to each comment is provided in the italicized text after each numbered item.

  1. Some writing typo like page 3, line 123 "traditional."

Fixed, thanks!

  1. Please change Figure 1 and using your device image to replace it.

Thanks for this suggestion.  Changing the image in Figure 1 would change the point we’re trying to make with the image.  In this image, we are showing what a traditional adhesive electrode looks like and the skin discomfort it can cause.  The images of our woven electrodes are available in Figure 3.

  1. How did the author obtain table 1? Is it just a survey for the volunteer? If so, I do not think this table is appropriate since the sample size is too small.

This table shows median scores for each of the characteristics listed in the comfort survey.  Because of our small sample size, we do not attempt to identify statistically significant differences between the electrode types but instead use this to illustrate specific features that are important when comparing the two.  We would like to keep this table as part of the manuscript to provide numerical references for the results covered in Section 5.3.

  1. Did the author consider the effect of sweat (will sweat affect the conductivity)? 

We did think about the variability that sweat can cause, and this is mentioned in line 275, but we did not measure changes in sweat during testing.  We have added a statement to this effect in Section 5.2.

  1. It will be more convincible if the author can provide a plot figure (similar to Figure 6) to evaluate the washability (before and after the wash). Also, a microscope image before and after washing is necessary.

We agree that it would be great to have a sample of the signals and microscopic images of the electrodes before and after washing, but we were not able to conduct this testing and therefore cannot add it to the manuscript. 

Round 2

Reviewer 2 Report

No more questions. Congratulations.